

# Generation of human liver organoids from pluripotent stem cell-derived hepatic endoderms

Kasem Kulkeaw[1], Alisa Tubsuwan[2], Nongnat Tongkrajang[1] and Narisara Whangviboonkij[1]

[1] Department of Parasitology, Faculty of Medicine Siriraj Hospital, Mahidol University, Bangkok, Thailand
[2] Stem Cell Research Group, Institute of Molecular Biosciences, Mahidol University, Nakhon Pathom, Thailand

Corresponding author
Kasem Kulkeaw,
kasem.kuk@mahidol.edu

## ABSTRACT

**Background**. The use of a personalized liver organoid derived from human-induced pluripotent stem cells (HuiPSCs) is advancing the use of in vitro disease models for the design of specific, effective therapies for individuals. Collecting patient peripheral blood cells for HuiPSC generation is preferable because it is less invasive; however, the capability of blood cell-derived HuiPSCs for hepatic differentiation and liver organoid formation remains uncertain. Moreover, the currently available methods for liver organoid formation require a multistep process of cell differentiation or a combination of hepatic endodermal, endothelial and mesenchymal cells, which is a major hurdle for the application of personalized liver organoids in high-throughput testing of drug toxicity and safety. To demonstrate the capability of blood cell-derived HuiPSCs for liver organoid formation without support from endothelial and mesenchymal cells.

**Methods**. The peripheral blood-derived HuiPSCs first differentiated into hepatic endoderm (HE) in two-dimensional (2D) culture on Matrigel-coated plates under hypoxia for 10 days. The HE was then collected and cultured in 3D culture using 50% Matrigel under ambient oxygen. The maturation of hepatocytes was further induced by adding hepatocyte growth medium containing HGF and oncostatin M on top of the 3D culture and incubating the culture for an additional 12–17 days. The function of the liver organoids was assessed using expression analysis of hepatocyte-specific gene and proteins. Albumin (ALB) synthesis, glycogen and lipid storage, and metabolism of indocyanine were evaluated. The spatial distribution of albumin was examined using immunofluorescence and confocal microscopy.

**Results**. CD34+ hematopoietic cell-derived HuiPSCs were capable of differentiating into definitive endoderm expressing *SOX17* and *FOXA2*, hepatic endoderm expressing *FOXA2*, hepatoblasts expressing *AFP* and hepatocytes expressing *ALB*. On day 25 of the 2D culture, cells expressed *SOX17*, *FOXA2*, *AFP* and *ALB*, indicating the presence of cellular heterogeneity. In contrast, the hepatic endoderm spontaneously formed a spherical, hollow structure in a 3D culture of 50% Matrigel, whereas hepatoblasts and hepatocytes could not form. Microscopic observation showed a single layer of polygonal-shaped cells arranged in a 3D structure. The hepatic endoderm-derived organoid synthesis ALB at a higher level than the 2D culture but did not express definitive endoderm-specific *SOX17*, indicating the greater maturity of the hepatocytes in the liver organoids. Confocal microscopic images and quantitative ELISA confirmed

albumin synthesis in the cytoplasm of the liver organoid and its secretion. Overall, 3D culture of the hepatic endoderm is a relatively fast, simple, and less laborious way to generate liver organoids from HuiPSCs that is more physiologically relevant than 2D culture.

## INTRODUCTION

Organoids are in vitro three-dimensionally (3D) self-organizing cells capable of imitating the structure and function of tissues or organs originating in the human body. Given their lower complexity and simpler experimental accessibility compared to tissues or organs in the body, organoids are useful for studying organ development (*Sato et al., 2009*), genetic diseases (*Dekkers et al., 2013*; *Huch et al., 2015*), cancers (*Boj et al., 2015*; *Gao et al., 2014*; *Hubert et al., 2016*), and host–pathogen interactions (*Bartfeld et al., 2015*; *Castellanos-Gonzalez et al., 2013*; *Farin et al., 2014*; *Zomer-van Ommen et al., 2016*) under a physiologically relevant microenvironment. However, to generate an organoid, cells or parts of tissue need to be isolated from an embryo, fetus, or adult organs. Such an invasive procedure has led to the decreased accessibility of adult human samples. Moreover, the limited availability and ethical concerns related to human embryonic and fetal tissues additionally restrict their use. Given advances in stem cell technology, human-induced pluripotent stem cells (HuiPSCs) are an alternative source for generating organoids for studying human brain development (*Lancaster et al., 2013*), polycystic kidney disease (*Xia et al., 2013*) and infectious diseases (*Cugola et al., 2016*; *Garcez et al., 2016*; *Finkbeiner et al., 2012*; *Forbester et al., 2015*; *Leslie et al., 2015*).

Owing to the physiological relevance of organoids, the use of patient-derived organoids has become a useful tool to customize a specific treatment for an individual patient, which is known as personalized medicine (*Mun et al., 2019*). To apply HuiPSC-derived organoids for personalized medicine, HuiPSCs first need to be generated from a patient's somatic cells. Skin fibroblasts are a primary source for HuiPSC generation, but this involves invasive biopsy of skin (*Mun et al., 2019*). Second, to form a functioning liver organoid, HuiPSCs must further differentiate into different cell populations, such as the hepatic endoderm, endothelium, and septum mesenchyme (*Takebe et al., 2012*; *Takebe et al., 2013*). Collectively, current methods are complicated and time consuming, limiting their use to a high-throughput setting (*Takebe et al., 2013*). Here, we developed a simpler method in which the hepatic endoderm alone was able to form liver organoids without coculture with the endothelium and septum mesenchyme. Moreover, this report also demonstrates that hematopoietic progenitor-derived HuiPSCs are capable of hepatocyte differentiation and liver organoid formation, highlighting the possibility for the use of a less invasive procedure to generate HuiPSCs.

## MATERIALS & METHODS

### Ethical approval/declaration

The study research protocol for the use of MUi019 was approved by the Mahidol University Central Institutional Review Board (COA no. 2016/038.2803) and the Human Research Protection Unit, Faculty of Medicine Siriraj Hospital, Mahidol University (COA no. Si459/2019).

### Culture of human iPS cells (*Si-Tayeb et al., 2010*)

HuiPS cells (MUi019 line) (*Tangprasittipap et al., 2017*) were cultured on a cell culture plate coated with 2 µg/mL Matrigel (Growth Factor Reduced; Corning, BD Bioscience). The culture medium for iPS cells consisted of Dulbecco's modified Eagle medium (DMEM) F12, 100 µM penicillin/streptomycin (Invitrogen), L-glutamine, 100 ng/mL fibroblast growth factor 2 (FGF2 or bFGF; Invitrogen), two ng/mL TGFβ1, 64 µg/mL L-ascorbic acid, and 1X insulin/transferrin/selenium. Cells were passaged every 5–6 days using 0.5 mM EDTA (Invitrogen) as the cell dissociation solution.

### Generation of hepatic endoderm, hepatoblasts, and hepatocytes from human iPS cells (*Si-Tayeb et al., 2010*)

HuiPS cells were cultured as mentioned above for 5–6 days. Hepatic differentiation consisted of four sequential phases. For the first phase, the culture medium consisted of 100 ng/mL activin A (R&D Systems) in RPMI plus 1x B27 (RPMI/B27) medium (Invitrogen) under ambient oxygen/5% $CO_2$ for 5 days. For the second phase, the culture medium contained 20 ng/mL bone morphogenetic protein 4 (BMP4, Peprotech) and 10 ng/mL basic FGF (Invitrogen) in RPMI/B27 medium under 4% $O_2$/5% $CO_2$ for 5 days. For the third phase, the culture medium consisted of 20 ng/mL hepatic growth factor (HGF, Peprotech) in RPMI/B27 medium under 4% $O_2$/5% $CO_2$ for 5 days. For the final phase, the cells were cultured with 20 ng/mL oncostatin-M (R&D Systems) in RPMI/B27 medium for 5 days and then maintained in ambient oxygen/5% $CO_2$.

### Gene expression analysis

RNA was extracted, and cDNA was prepared using a cDNA synthesis kit (Biotechrabbit). The expression of pluripotency-related, endodermal and hepatic genes was assessed using real-time conventional PCR. Primer sets were obtained from previous reports (*Schwartz et al., 2002*; *Takahashi et al., 2007*; *Chen et al., 2012*; *Irie et al., 2015*; *Aguila et al., 2014*) and are shown in Table 1. Luna® Universal qPCR Master Mix (New England BioLabs) was used, and the primer concentration was 1 µM for each primer. The thermal cycles were as follows: initial denaturation at 95 °C for 3 min, 30 cycles of denaturation at 95 °C for 10 s, annealing at 60 °C for 10 s, and extension at 72 °C for 10 s, followed by a final extension at 72 °C for 1 min. Transcript of human actin beta (*ACTB*) served as internal control to normalize gene expression. Threshold cycles (CT) of each samples were compared with a reference sample using the $2^{-\Delta\Delta CT}$ method (*Rao et al., 2013*) and is shown as relative expression. Gene expression analyses were from three independent experiments, and each run of real-time PCR were carried out in triplicate.

**Table 1** Primer sequences used for quantitative PCR in the study.

| Genes | Primers (5′–3′) |
|---|---|
| OCT4 | Fw: GCTGGAGCAAAACCCGGAGG |
| | Rv: TCGGCCTGTGTATATCCCAGGGTG |
| SOX17 | Fw: GAGCCAAGGGCGAGTCCCGTA |
| | Rv: CCTTCCACGACTTGCCCAGCAT |
| FOXA2 | Fw: TATGCTGGGAGCGGTG |
| | Rv: TGTACGTGTTCATGCCGTTCA |
| HNF4 | Fw: GGCAATGTGTCAGGGAGGAA |
| | Rv: CAGGGATTTCAGGGGCACTT |
| AFP | Fw: GCAGCCAAAGTGAAGAGG |
| | Rv: TGTTGCTGCCTTTGTTTG |
| ALB | Fw: AGACAAATTATGCACAGTTG |
| | Rv: TTCCCTTCATCCCGAAGTTC |

## Immunofluorescence

The cells or organoids were attached to a glass slide and air-dried. The cells were fixed with 1% paraformaldehyde in PBS at room temperature for 30 min and then permeabilized with 0.05% Triton X-100 in PBS for 15 min. After washing with PBS, the cells were exposed to 3% BSA in PBS for 15 min. Rabbit polyclonal anti-human HNF4 (1:100 from Sigma-Aldrich), mouse monoclonal anti-human α-fetoprotein antibody (1:500, Sigma-Aldrich), rabbit polyclonal anti-human albumin (1:100, Sigma-Aldrich), rabbit polyclonal anti-human CYP3A43 antibody (1:100, Abcam), mouse monoclonal anti-human EpCAM antibody (1:100, GeneTex), rabbit polyclonal anti-human CD31 antibody (1:20, Abcam) and mouse anti-human CD81 (1:200, Abcam) antibodies diluted in 1% BSA were applied onto the glass slide and incubated overnight. After washing with PBS, the cells were then incubated with Alexa Fluor 594-conjugated goat anti-rabbit IgG and Alexa Fluor 488-conjugated goat anti-mouse IgG antibodies (Invitrogen). The DAPI-containing medium was mounted on the glass slide and covered with a glass coverslip. Images of the cells were visualized using a confocal microscope (ECLIPSE Ti-Clsi4 Laser Unit, Nikkon).

## Generation of liver organoids from the hepatic endoderm

HuiPSC-derived hepatic endoderm was collected from a well of a 24-well plate using cell dissociation buffer (Gibco, LifeTechnologies). The cells were then suspended in a culture medium consisting of 40 ng/mL HGF and 20 ng/mL oncostatin M in RPMI/B27. The cell suspension was immersed in ice for 15 min and immediately mixed with an equal volume of Matrigel. The final mixture contained 50% Matrigel, 20 ng/mL HGF (Peprotech), and 10 ng/mL oncostatin M. After solidification of the Matrigel at 37 °C for 15 min, RPMI/B27 medium containing 20 ng/mL HGF and 10 ng/mL oncostatin M was added on top of the solid Matrigel. The cells were cultured under 4% $O_2$/5% $CO_2$ at 37 °C. The covering medium was renewed every 48 h. From day 10 onward of 3D culture, the medium on top of Matrigel was changed to the HCM$^{TM}$ Hepatocyte Culture Medium BulletKit$^{TM}$ (herein called HCM) containing HBM$^{TM}$ Basal medium and HCM$^{TM}$ SingleQuots Supplements, which include transferrin, ascorbic acid, human endothelial growth factor, insulin, hydrocortisone, fatty
acid-free BSA and Gentamicin sulfate-Amphotericin. To sub-culture, the plate was place on ice for 10 min. The cold HCM was then added into a well and gently mixed by pipetting to liquefy the Matrigel. A part of liquefied Matrigel was suspended in the cold HCM. An equal volume of the organoid mixture was then mixed with Matrigel and plated into a 96-well plate. After solidification of the Matrigel at 37 °C for 15 min, the HCM was added on top. The HCM was changed every 48 h.

## Enzyme-linked immunosorbent assay (ELISA) for human albumin

Amount of human albumin secreted in culture medium was assessed using the Human Albumin/Serum albumin ELISA kit (Millipore) following manufacturer's instruction. The hepatocyte culture medium was used as background control. Triplicates were performed for each independent experiments.

## Glycogen accumulation

The cells were treated with 4% paraformaldehyde in PBS and permeabilized with 0.5% Triton X-100 in PBS. Cells treated with 1 mg/mL diastase in PBS (Sigma) were used as the negative control. The cells were then incubated with periodic acid for 5 min, washed with distilled water, and incubated with freshly prepared Schiff's solution for 15 min. Finally, the cells were rinsed, and the nuclei were stained with hematoxylin. After rinse with water, cells were incubated with Bluing reagent for 30 s followed by incubation with Light Green solution for two min. The glass slide was immersed in absolute alcohol to dehydrate and air-dried.

## Lipid stain using Oil Red O

Lipid stored in cells was visualized using fat-soluble Oil Red O following a manufacture' instruction (Oil Red O Stain Kit, Abcam). Cells were attached onto a glass slide using cytocentrifugation. After air dry, the glass slide was placed in propylene glycol for 2 min and then incubated with Oil Red O solution for 6 min. The glass slide was immersed in 85% propylene glycol in distilled water for 1 min and rinsed twice with distilled water. The cells were then stained with Hematoxylin for 1–2 min and rinse thoroughly in tap water. Then, the glass slide was rinsed with two changes of distilled water and air-dried.

## Metabolism of indocyanine

Indocyanine green (ICG) is an organic anion, non-toxic and exclusively eliminated by hepatocytes and has been useful as a marker of hepatocyte. The ICG was added into culture medium at 1 mg/mL and incubated at 37 °C for 60 min. After medium removal, cells were washed with PBS to remove excess extracellular indocyanine.

## Statistical analysis

Data are presented as means ± standard deviation (SD). Differences were statistically evaluated using Student t test. $P$-values <0.05 were considered statistically significant.

## RESULTS

### Hepatic differentiation potential of the HuiPS MUi019 cell line

The potential of the peripheral blood CD34+ hematopoietic progenitor cell-derived MUi019 cell line for endodermal differentiation has been demonstrated in vivo (*Tangprasittipap et al., 2017*); however, its capability for hepatic differentiation in vitro is unknown. Here, we first assessed the capability of the HuiPSC MUi019 cell line (*Tangprasittipap et al., 2017*) for hepatic differentiation. Hierarchically differentiated cells, namely, definitive endoderm, hepatic endoderm (HE), hepatoblasts, and hepatocytes, were prepared according to a method described in a previous report (*Si-Tayeb et al., 2010*). Upon exposure to four different sets of cytokines (Fig. 1A), the morphologies of the differentiated cells tended to change in a stepwise fashion (Figs. 1B–1E). On day 25 of differentiation, groups of cells having the same morphology appeared in many focal areas of the cell culture plate (Figs. 1E–1F). Higher magnification observation revealed polygonal cells having large nuclei, which is a main characteristic of hepatocytes (Fig. 1F).

To characterize the hepatic cells, mRNA transcripts indicating each cell type were examined using real-time PCR. Here, octamer-binding transcription factor 4 (*OCT4*) identified an undifferentiated pluripotent stem cells, forkhead box A2 (*FOXA2*) and sex determining region Y box 17 (*SOX17*) indicated definitive endoderm. The hepatic endoderm expresses *FOXA2*. Alpha-fetoprotein (*AFP*) and albumin (*ALB*) indicated hepatoblast and mature hepatocytes, respectively. Based on their transcript profile in Figs. 1G–1K, mRNA level of *OCT4* decreased as the HuiPSC differentiated into hepatocytes (Fig. 1G). By contrast, that of *SOX17* and *FOXA2* up-regulated in the 10-day 2D culture, in which the HuiPSCs specified to hepatic endoderm, respectively; however, level of both transcripts gradually decreased in the 15-day 2D culture during differentiation of hepatic endoderm into hepatoblasts (Figs. 1H and 1I, respectively). Level of *AFP* transcript was detectable at day 10 until day 25 (Fig. 1J), while that of *ALB* transcript could be detected at day 25 of 2D culture (Fig. 1K). Collectively, the 25-day-old cells exhibited characteristics of endoderm (*SOX17* and *FOXA2*), hepatoblasts (*AFP*), and hepatocyte (*ALB*) cells. In Figs. 1L–1Q, confocal microscopic observations confirmed the presence of intracellularly synthesized albumin (Fig. 1L), a specific marker of hepatic cells, and CD81 (Fig. 1M), an important receptor of the *Plasmodium falciparum* sporozoite (*Silvie et al., 2003*) and hepatitis C virus entry (*Bruening et al., 2018*). Coexistence of albumin and CD81 was also observed (arrowheads in Fig. 1O). Cells stained with 2nd antibody was served as background control (Fig. 1P). Representative confocal image revealed 5.9 ± 1.3% of the 25-day-old 2D cultured cells expressing albumin (Fig. 1Q). The HuiPSC-derived hepatocytes were able to store glycogen (Fig. 1R). Taken together, the results indicated that the HuiPS MUi019 cell line was capable of hepatic cell differentiation in vitro.

### HuiPSC-derived hepatic endoderm spontaneously formed organoids, but the hepatoblasts and hepatocytes could not

To determine which cell type is capable of forming an organoid, we collected HE, hepatoblasts, and hepatocytes at days 10, 15, and 20 of cell differentiation (Fig. 2A) and cultured them in a 3D setting (50% Matrigel prepared in RPMI supplemented with 1%
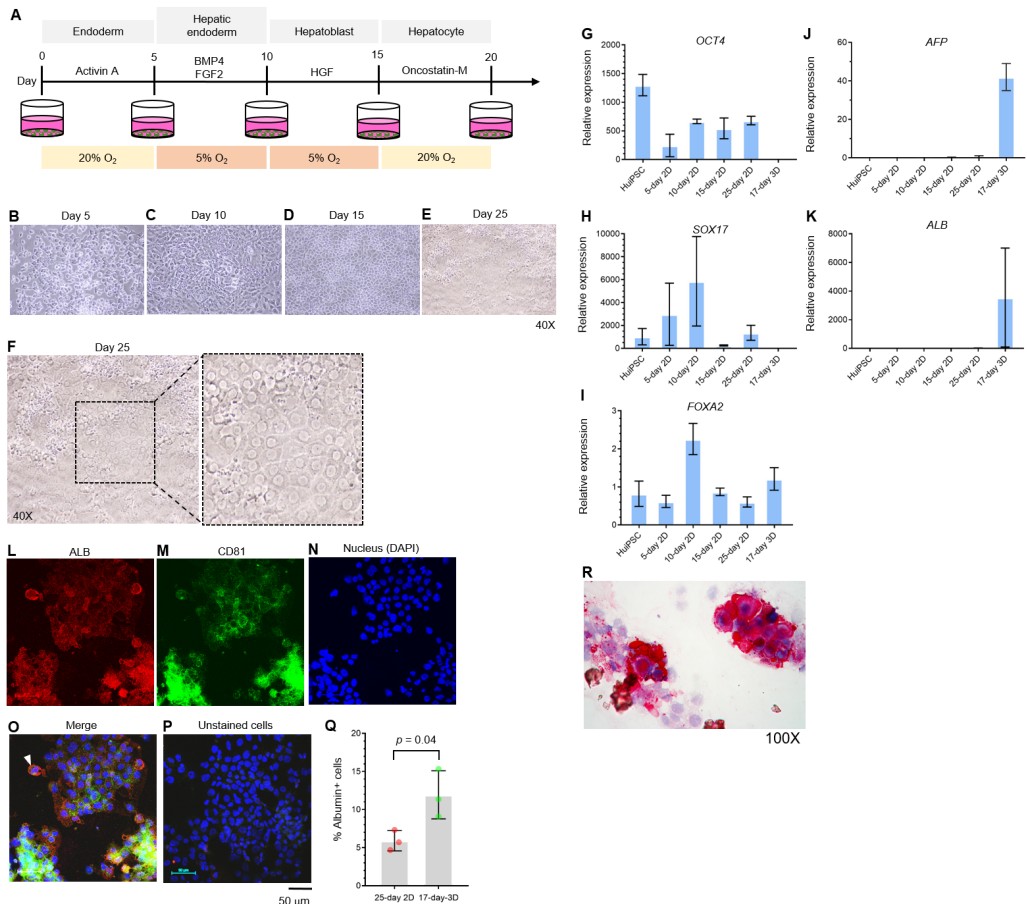

**Figure 1** **Hepatic differentiation potential of the HuiPSC MUi019 line.** (A) Schematic diagram of hepatic cell differentiation in the 2D culture system. The protocol consisted of four stepwise phases: endoderm, hepatic endoderm, hepatoblast, and hepatocyte. Cells were cultured under 5% or 20% $O_2$. (B–E) Representative microscopic bright-field images of differentiated iPS cells after exposure to different sets of cytokines (20X objective lens). (F) Microscopic image of 25-day-old differentiated cells at higher magnification. The dotted line indicates the area where polygonal cells having large nuclei were observed. (G–K) Gene expression profile of cells at different stages of culture. Relative expression levels were calculated using the $2^{-\Delta\Delta CT}$ method. Data are the mean ± SD ($n = 3$) and statistically analyzed using Student's $t$ test. (L–P) Confocal microscopic observation of albumin (red color) and CD81 (green color). Scale bar = 50 mm. (Q) Percentage of albumin-expressing cells observed using immunofluorescence and confocal microscopy. (R) Glycogen storage in the 25-day differentiated cells from E. Representative image shown was obtained with a 100X objective lens. Light microscopic and confocal images are representative of three independent experiments.

B27, HGF, and oncostatin M) (Fig. 2B). The top of the solidified Matrigel was covered with the same medium described above without Matrigel. On day 5 of the 3D culture, a small, irregularly shaped structure appeared in the culture of the HEs (Fig. 2C), while neither hepatoblasts nor hepatocytes were observed (Figs. 2D and 2E, respectively). Therefore, 2D-cultured, 10-day-old HE was used further for organoid formation. On day 7 of the 3D culture, irregularly shaped cell clumps composed of an outer layer and inner mass could be clearly observed under a microscope. The outer layer had a protruding, spike-like structure

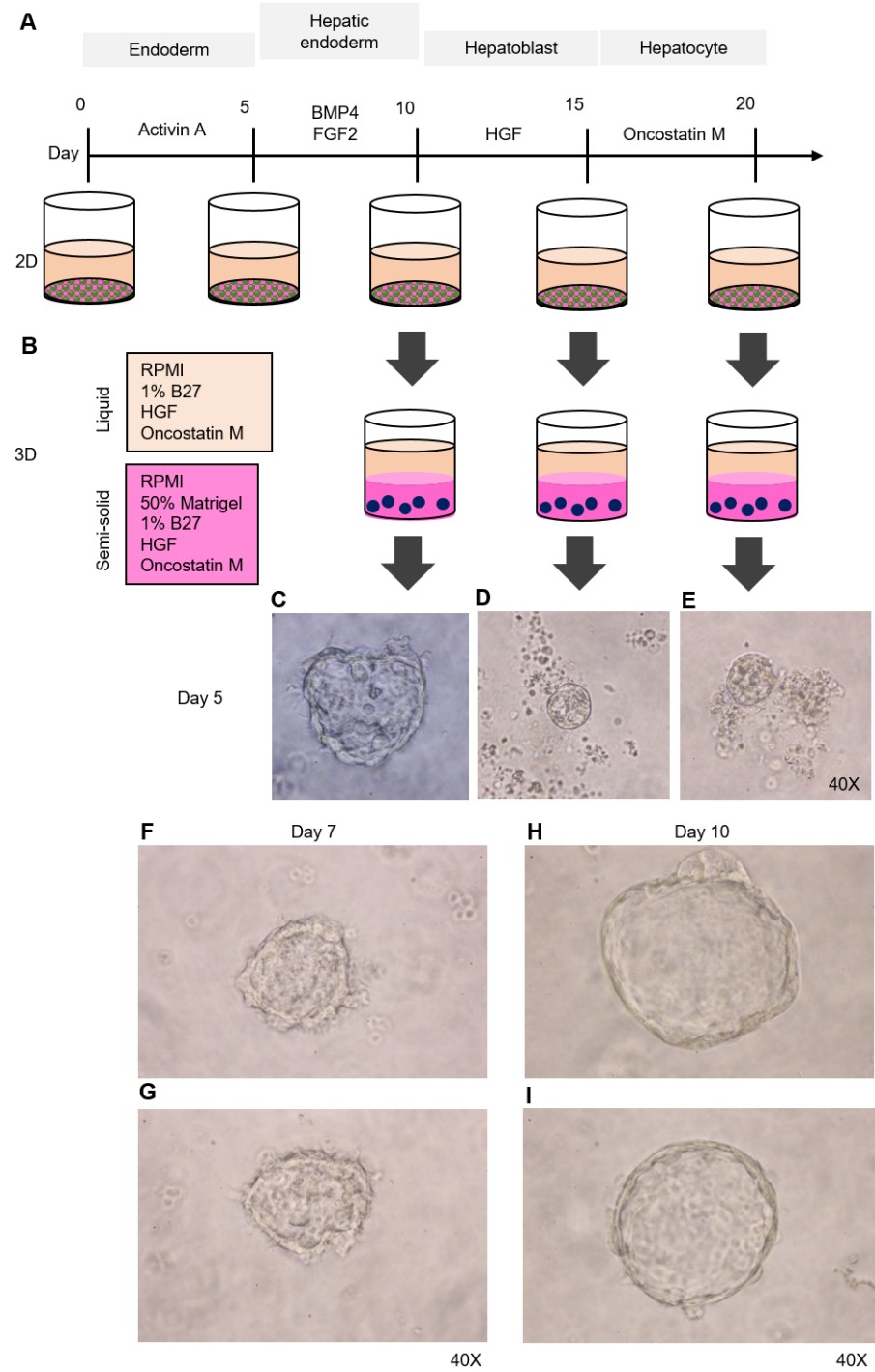

**Figure 2  Organoid formation potential of the hepatic endoderm, hepatoblast, and hepatocyte phases.**
(A) Schematic diagram of the methods used for assessing organoid formation ability. (B) Hepatic endoderm (day 10), hepatoblasts (day 15), and hepatocytes (day 20) were cultured in semisolid Matrigel (pink color). Liquid medium (light orange) was added on top of the Matrigel. (C–E) Ability of hepatic endoderm, hepatoblasts, and hepatocytes to form an organoid in the Matrigel-based 3D culture was assessed at day 5 of the 3D culture. Images shown were obtained with a 40× objective lens. (F–G) Two representative images of the organoid in the Matrigel-based 3D culture at day 7. (H–I) Two representative images of the organoid in the Matrigel-based 3D culture at day 10. Three independent experiments were performed and representative microscopic images are shown.

(Figs. 2F–2G). By day 10 of the 3D culture, the sizes of cell clumps increased, while their pikes disappeared, and the outer layer became smooth (Figs. 2H–2I). At this time point, the organoid showed the characteristics of a spherical, hollow shape, hereafter called a hepatic endoderm organoid or HEO. From day 10 onward of the 3D culture, the covering medium was changed to the HCM for hepatocyte maturation.

## Characterization of the HuiPS-derived HEOs

On day 12 of the 3D culture, the HEO had space at the center surrounded by a layer of cells, which are characteristics typical of a blastocyst (Fig. 3A). The sizes of the HEOs varied and increased from day 12 onward (Figs. 3B–3C). Despite their increasing sizes, there was no fusion of adjacent HEOs. Instead, the outer layer of the HEO was bent (Figs. 3B and 3C, arrowheads). Microscopic observation of the HEOs on days 12 and 14 revealed a thin, single-layer structure (Figs. 3D and 3E, respectively). By day 17, dome-shaped cells appeared (Fig. 3F, arrowheads). Higher magnification observation of the 17-day HEO yielded microscopic images showing a very thin layer (Fig. 3G). When focusing on the cells, polygonal cells could be observed (arrowheads in Fig. 3H).

To characterize the 17-day-old HEOs, gene and protein expression analyses were performed. Although the HEOs did express the pluripotency-related *OCT4* (Fig. 1G) and the endoderm-specific transcription factor *SOX17* (Fig. 1H); however, the level was much lower than the 2D culture. By contrast, the 17-day-old HEOs expressed hepatic transcription factor *FOXA2* (Fig. 1I), hepatoblast-specific factor *AFP* (Fig. 1J), and hepatocyte-specific *ALB* (Fig. 1K) at higher level compaired to the 25-day-old 2D-cultured hepatocyte, thus implying the greater maturation of the 3D culture. Then, we evaluated the expression of hepatocyte-related proteins in a time course manner using immunofluorescence and confocal microscopy. The confocal images revealed expression of hepatic nuclear factor 4 (HNF4) at both day 12 (Figs. 3I–3K) and 17 (Figs. 3L–3N) of the 3D culture. Moreover, the confocal images also confirmed that the *ALB* transcripts were translated into albumin (Figs. 3O–3T). As shown in Figs. 3U–3Z, the HEOs expressed AFP at day 12 and 17 of the 3D culture. Although CYP4A3 was not observed at the 12-day-old HEOs, CYP4A3 was subsequently detected at day 17 of the 3D culture (Figs. 3AA–3FF). To ensure that HEO could be applied as a model of infectious diseases, we examined the expression of CD81, an important receptor for *Plasmodium falciparum* sporozoites and hepatitis C virus. As shown in Figs. 3GG–3LL, the 12-day- and 17-day-old HEOs expressed CD81. Localization of albumin and CD81 was performed. Albumin was expressed in an asymmetric pattern (Figs. 3MM–3PP); however, the expression of CD81 overlapped with that of albumin in some areas of the HEOs (Fig. 3PP). Higher magnification images revealed the coexpression of albumin and CD81 (Figs. 3QQ–3TT). Based on cell counting, higher percentage of ALB+ cells was observed in the HEOs (bar graph in Fig. 1Q). All confocal images were compared with the HEOs stained with the 2nd antibody (Figs. 3UU–3XX). In comparable to immunofluorescence, the 17-day-old HEOs secreted albumin into the culture medium at $1,077 \pm 176$ ng per $10^6$ examined cells within 48 h of culture (Fig. 3YY). By contrast, among three independent experiments, the 25-day-old 2D cultured hepatic cells secreted albumin into the culture medium at $44.9 \pm 53$ ng per $10^6$ cells within 48 h of culture.

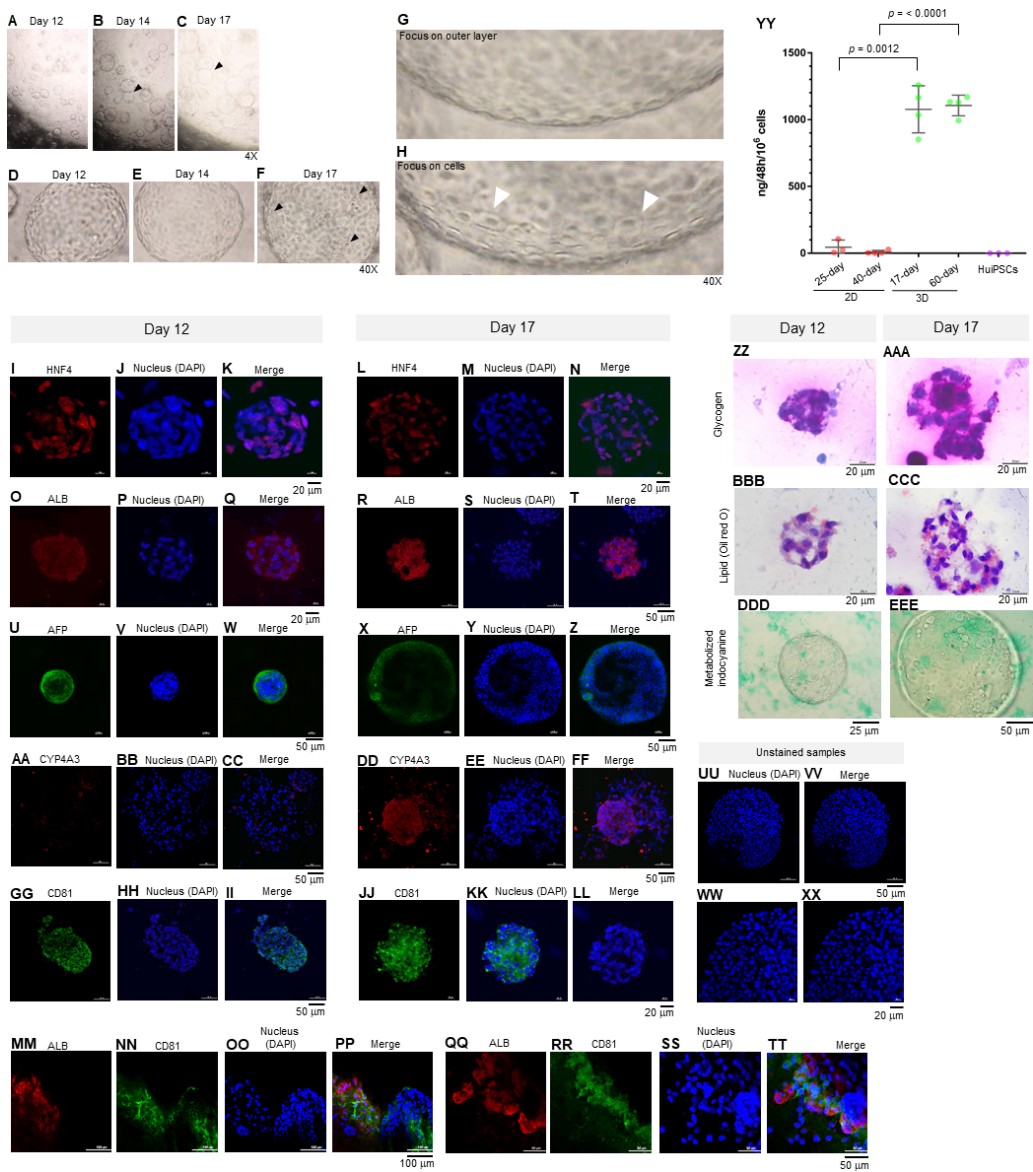

**Figure 3** **Morphology of hepatic endoderm-derived organoids (HEOs).** (A–C) Morphology of the HEOs at days 12, 14, and 17 postculture in Matrigel (4X objective lens). (D–F) Representative images of single HEOs are shown at high magnification (40X). (G) The outer layer of cells is shown. (H) Zoomed-in image of the HEOs showing the cells with a polygonal shape and large nuclei (arrowheads). (I–LL) Confocal images of a day 12- and 17-derived organoid showing the expression of hepatic nuclear factor 4 (HNF4), hepatocyte-specific albumin (ALB) and $\alpha$-fetoprotein (AFP), cytochrome P450 4A3 (CYP4A3), and CD81, an important receptor of *Plasmodium falciparum* sporozoites. (MM–PP) Co-localization of CD81 and ALB in the 17-day HEO. The confocal images of organoid show the expression of CD81 (green) and hepatocyte-specific albumin (red). (QQ–TT) Zoomed-in images reveal the albumin- and CD81-expressing cells. (UU–XX) Cells stained with 2nd antibody specific to IgG of rabbit and mouse (unstained samples) served as background control. Four independent experiments were performed and representative images are shown. (YY) Total amount of human albumin secreted within 48 h in the culture medium of the 25- and 40-day 2D culture and the 17- and 60-day 3D culture. Total amount (ng) of human albumin in each experiments were calculated based on number of cells. Individuals represent independent experiments. Data are the mean ± SD ($n$ =3–4) and statistically analyzed using Student's t test. Culture medium of undifferentiated HuiPSCs was used for comparison. (ZZ–EEE) Hepatocyte functions (glycogen and lipid storage, and metabolism of indocyanine) of a day 12- and 17-derived organoids. Four independent experiments were performed and representative images are shown.

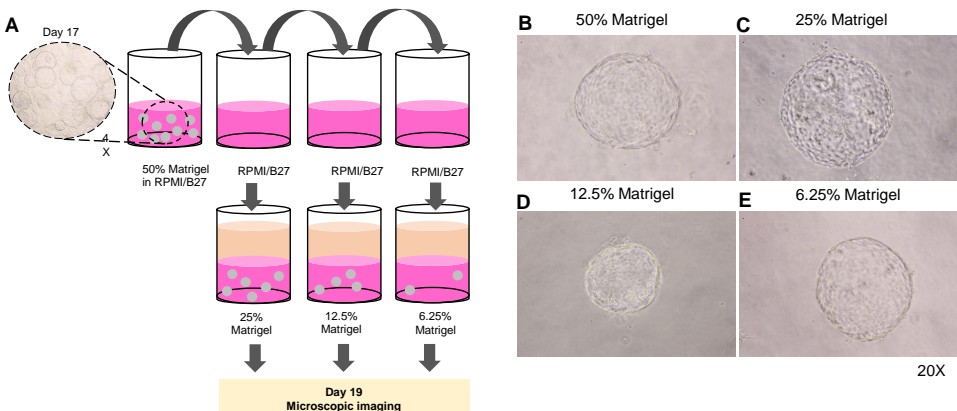

**Figure 4  Minimal concentration of Matrigel needed to maintain the 3D structure of the organoid.**
(A) Schematic diagram showing a method used to determine the minimum concentration of Matrigel in which the 3D structure of the organoid could be maintained. (B–E) Microscopic images of the liver organoid. The 17-day-old organoids were split to obtain 25%, 12.5%, and 6.25% Matrigel in RPMI/B27, which were supplemented with HGF and oncostatin M, and cultured for 5 days. All images were captured using a 20X objective lens. Three independent experiments were performed and representative images are shown.

Apart from the immunofluorescence, both day 12 and 17-derived HEOs were capable of storing glycogen (Figs. 3ZZ and 3AAA) and lipid (Figs. 3BBB–3CCC) and metabolizing indocyanine (Figs. 3DDD–3EEE) in a time-course manner. Collectively, the data of gene and protein expression imply a full maturation of the HEOs within 17 days of 3D culture, which is equivalent to the 27-day period.

## Minimal concentration of Matrigel required for maintaining the 3D structure of the HEOs

Given the high cost of Matrigel, we next determined the minimum concentration of Matrigel in which the 3D structure of the organoid remained unchanged. The 17-day-old organoids were split by dilution with the medium to obtain 25%, 12.5%, and 6.25% Matrigel prepared in RPMI/B27 supplemented with HGF and oncostatin M (Fig. 4A). Regardless of the Matrigel concentration, the HEOs could maintain their spherical structure, and their outer layer remained intact (Figs. 4B–E). In our experience, using Matrigel at a concentration lower than 6.25% leads to difficulty in changing the upper liquid medium during long-term culture.

## Long-term culture of the HEOs

We further evaluated whether the HEOs could be maintained hepatocyte characteristics in a long-term culture. The HEOs were sub-cultured weekly and the culture continued for 60 days in total. Representative images of a well of 96-well plate show morphology and size of the 60-day-old HEOs under a microscope after a week of sub-culture (Fig. 5A). At higher magnification, the HEOs remain single layer of cells (Fig. 5B) having a polygonal shape plus large nuclei as well as binucleated cell (respective white and black arrowhead in Fig. 5C), characteristics of hepatic cells. Confocal images of the day 60-derived HEOs

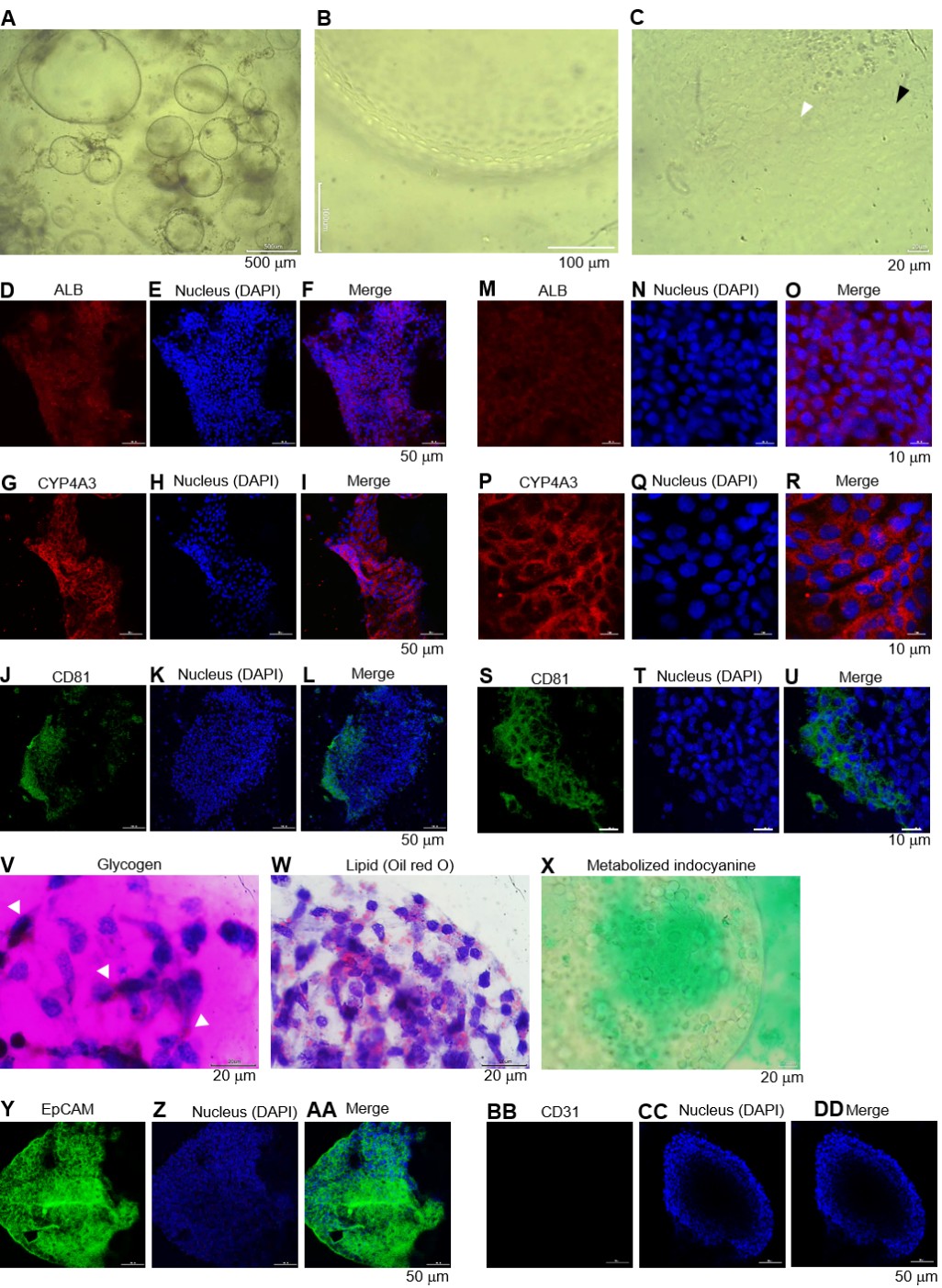

**Figure 5** **Long-term maintenance of the hepatic endoderm-derived organoid.** (A) Morphology of the HEOs at days 60 postculture in Matrigel. Representative images of a well of 96-well plate show the HEOs after a week of sub-culture. (B) At higher magnification, the outer layer of cells remain single. (C) Focused image of the HEOs showing the cells with a polygonal shape and large nuclei (white arrowhead) as well as binucleated cell (black arrowhead), characteristics of hepatic cells. 

**Figure 5 (…continued)**
(D–L) Confocal images of the day 60-derived HEOs showing the expression of hepatocyte-specific albumin (ALB), cytochrome P450 3A4 (CYP4A3) and CD81. (M–U) Zoomed-in confocal images shown in (D). (V–X) Hepatocyte functions (glycogen and lipid storage, and metabolism of indocyanine) of the day 60-derived organoids. Four independent experiments were performed and representative images are shown. (Y–AA) Confocal images of the day 60-derived HEOs showing the expression of EpCAM. (BB–DD) Confocal images of the day 60-derived HEOs showing the expression of CD31.

showing the expression of hepatocyte-specific ALB (Figs. 5D–5F), CYP4A3 (Figs. 5G–5I) and CD81 (Figs. 5J–5L). To clearly demonstrate, higher magnifications of the images are shown in Figs. 5M–5U. In agreement with immunofluorescence, the 60-day-old HEOs secreted albumin at $1,106 \pm 77$ ng per $10^6$ cells within 48 h, a similar level to the 17-day-old HEOs (Fig. 3YY). By contrast, when we maintained the 2D culture of the HuiPSC-derived hepatic cells for 40 days, lower amount of albumin could be detected relative to that of the day 25-derived hepatic cells (Fig. 3YY), implying loss of albumin synthesis an in vitro culture. Moreover, the day 60-derived HEOs were still able to storing glycogen (Fig. 5V) and lipid (Fig. 5W), and metabolizing indocyanine (Fig. 5X) as well as expressing hepatic stem/progenitor marker EpCM (Figs. 5Y–5AA) but not endothelial marker CD31 (Figs. 5BB–5DD). Taken together, the HEOs remained characteristics of hepatic cells after long-term culture in a Matrigel.

## DISCUSSION

Here, we developed a simple and fast protocol to generate a functioning human liver organoid from pluripotent stem cells derived from peripheral blood CD34+ cells. By day 15 of the initial culture of HuiPSCs, the HEOs had developed and required an additional 2–3 days to synthesize albumin, an indicator of a functioning hepatocyte. To the best of our knowledge, the protocol used in this study is faster than that reported in a recent study (*Mun et al., 2019*) and does not require a mixture of endothelial cells and mesenchymal cells (*Takebe et al., 2012*; *Takebe et al., 2013*), endothelial cells (*Pettinato et al., 2019*), or hepatic progenitor cells (*Ng et al., 2018*) as starting components. In agreement with the results of other studies, the spherical structures of liver organoids were also reportedly observed (*Huch et al., 2015*; *Mun et al., 2019*).

Several studies have attempted to generate fully functioning liver organoids. Coculture of HuiPSCs with adipose microvascular endothelial cells in 3D culture could form liver organoids (*Pettinato et al., 2019*). iPS cell-derived hepatic progenitors reportedly formed liver organoids using an inverted colloid crystal, which provided mechanical properties to recapitulate the extracellular niche (*Ng et al., 2018*). Moreover, spherical structures emerged from a 2D culture monolayer of HuiPSCs that were capable of forming liver organoids in liquid and semisolid cultures; however, it took approximately 22 days to obtain the spherical structures, and additional 8–9 days as well as two stepwise medium were needed for maturation (*Mun et al., 2019*). Compared to the results of these reports, the advantages of our protocol, in which mature hepatocyte could be obtained by day 27 of HuiPSC culture, include simplicity and rapidness (*Takebe et al., 2013*; *Takebe et al., 2012*; *Ng et al., 2018*; *Pettinato et al., 2019*). Nevertheless, there remains a drawback in our protocol.

Given a lack of comparison with primary hepatocyte culture, how extent maturation of hepatocytes in the 27-day-old HEOs is unknown. Thus, we indirectly compared amount of secreted albumin in culture of primary hepatocyte (1,500–1,700 ng/mL/day/$10^6$ cells) (*Mun et al., 2019*) with the 27-day-old HEOs (1,077 $\pm$ 176 ng/48h/$10^6$ cells). The comparison implies that the 27-day-old HEOs were immature relative to the 2D culture of primary hepatocytes. In agreement, expression of EpCAM, which mature hepatocytes are devoid of (*Dolle et al., 2015*), was observed in most cells of the 27-day-old HEOs (Fig. 5AA). Therefore, lower extent of maturation was likely a limitation of our protocol.

iPS cells can be generated from various tissue sources, including fibroblasts from skin (*Alawad et al., 2016*; *Du et al., 2015*) and hematopoietic cells (*Haase, Gohring & Martin, 2017*; *Takenaka et al., 2010*; *Kim, Manzar & Zavazava, 2013*) or lymphocytes (*Phillips et al., 2012*) from peripheral blood, and have been used in disease modeling. HuiPSC lines derived from peripheral blood T lymphocytes reportedly differentiate into hepatocytes more efficiently than hiPSC clones derived from adult dermal fibroblasts (*Kajiwara et al., 2012*). To the best of our knowledge, this study is the first report showing that CD34+ cell-derived iPS cells could differentiate into hepatocytes and hepatic endoderm with the ability to give rise to liver organoids. Collecting peripheral blood is less invasive than skin biopsy; thus, PB has become the first choice for the generation of PSCs. Therefore, our protocol is probably suitable for specific and safe personalized therapy for hepatotropic diseases. To ensure that our protocol could be used to generate a model of hepatotropic diseases, we therefore examined the expression of the pathogen entry receptor. The iPS-derived liver organoids expressed CD81 (*Foquet et al., 2015*; *Silvie et al., 2006*; *Silvie et al., 2003*; *Yalaoui et al., 2008*), a potent cell receptor allowing *Plasmodium* sporozoite invasion into hepatocytes. Thus, this would allow the examination of host–pathogen interactions in a model that is physiologically relevant to the human body. Regarding the clinical application of liver organoids in transplantation, it is necessary to demonstrate further that the HEOs generated from this study are capable of rescuing animals affected by lethal hepatic diseases.

## CONCLUSIONS

CD34+ cell-derived HuiPSCs were capable of differentiating into hepatocytes and hepatic endoderm with the ability to give rise to liver organoids. 3D culture of the hepatic endoderm is a relatively fast, simple, and less laborious way to generate liver organoids from HuiPSCs that is more physiologically relevant than 2D culture. Further examination of host–pathogen interactions in this HE-derived liver organoids is necessary.

## ACKNOWLEDGEMENTS

The authors gratefully acknowledge Prof. Dr. Wanpen Chaicumpa for valuable scientific suggestion, and the Institute of Molecular Biosciences, the Siriraj Central Research Facility and Department of Anatomy, Faculty of Medicine Siriraj Hospital, Mahidol University for technical support.

### Funding

This work was supported by the Research Career Development Grant from Thailand Science Research and Innovation (grant no. [IO] RSA6280102) and a grant from the Siriraj Research Fund, Faculty of Medicine Siriraj Hospital, Mahidol University, Bangkok, Thailand (grant no. [IO] R016233004). The funders had no role in study design, data collection and analysis, decision to publish, or preparation of the manuscript.

### Grant Disclosures

The following grant information was disclosed by the authors:
Thailand Science Research and Innovation: RSA6280102.
Siriraj Research Fund, Faculty of Medicine Siriraj Hospital.
Mahidol University, Bangkok, Thailand.

### Competing Interests

The authors declare there are no competing interests.

### Author Contributions

- Kasem Kulkeaw conceived and designed the experiments, performed the experiments, analyzed the data, prepared figures and/or tables, authored or reviewed drafts of the paper, and approved the final draft.
- Alisa Tubsuwan, Nongnat Tongkrajang and Narisara Whangviboonkij performed the experiments, analyzed the data, prepared figures and/or tables, and approved the final draft.

### Ethics

The following information was supplied relating to ethical approvals (i.e., approving body and any reference numbers):

The study research protocol for the use of MUi019 was approved by the Mahidol University Central Institutional Review Board (COA no. 2016/038.2803) and the Human Research Protection Unit, Faculty of Medicine Siriraj Hospital, Mahidol University (COA no. Si459/2019).

### Data Availability

The raw data are uncropped images used as parts of Figs. 1–4.

### Supplemental Information

Supplemental information for this article can be found online at http://dx.doi.org/10.7717/peerj.9968#supplemental-information.

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
