# Peer review of "Generation of human liver organoids from pluripotent stem cell-derived hepatic endoderms"

_PeerJ, doi:10.7717/peerj.9968_

## Round 0.1 · original submission · Major Revisions

Although according to the three reviewer comments, the manuscript does not meet the quality necessary to be published on PeerJ, I will reconsider a revised version of the manuscript if you will correctly address criticisms from Reviewers 2 and 3. For your convenience, all reviewers' comments are listed below.

Reviewer 1 ·

Basic reporting

The authors describe how peripheral blood-derived HuiPSCs can differentiated into hepatic endoderm.
It is basically a method paper with no further development or applications to address biological or molecular questions.
Most findings are morphological, while describing hepatic endoderm, hepatoblast and hepatocyte phases.
To make this kind of paper acceptable at least a single cell characterization or a geographical distribution of specific proteins by histochemistry or fluorescence should be done. what shown in fig 3 is not sufficient.

Experimental design

Very basic experimental design

Validity of the findings

I do not find convincing evidence of novelty or advancement of knowledge by this paper

Additional comments

I suggest to use the organoids to make functional studies with other cells or proteins, run single cells or matrisome analysis

Reviewer 2 ·

Basic reporting

look at general comments

Experimental design

look at general comments

Validity of the findings

look at general comments

Additional comments

The manuscript “Generation of human liver organoids from pluripotent stem cell-derived hepatic endoderms” by Kasem Kulkeaw et al. describes a method to generate liver organoid from human CD34+ hematopoietic cell-derived iPSCs. This issue is interest and update and the proposed technology gives the possibility, in the future, to generate human liver organoids with a less invasive procedure, i.e. a simple blood test. These findings are considered to be interesting and of potential clinical importance, the manuscript is well written and the data are clearly described. However, in my opinion, the authors show results that do not justify several of the conclusion that they reach.

General criticisms:
-In order to demonstrate that the generated organoids are differentiated, authors should evaluate liver marker expression at both mRNA and protein levels and/or by immunofluorescence in a quantitative manner and not only by standard PCR. In Figures 1E and 3D the authors should evaluate and provide the percentage of ALB positive cells calculated on many cell fields or by FACS, and this analysis should be accompanied by quantification of the albumin secreted in the medium. Also the decrease of staminality markers, such as OCT4, should be evaluated during the differentiation process. Accumulation of glucose and other metabolites should be evaluated during the differentiation process (in a time corse) in 3D cultures.
-The method developed by the authors is simpler and less laborious than those described so far because it provides the use of iPSCs differentiated in hepatic endoderm alone without co-culture with endothelium and mesenchyme. Anyway, the mesenchymal cells (fibroblasts and vascular endothelial cells) dynamically interact with differentiating hepatocyte cells to promote further differentiation and provide an extracellular microenvironment. In this regard, it would be interesting to evaluate whether the human hepatic organoids generated by the authors maintain their mature hepatic characteristics over long-term culture. Also discuss this aspect in the discussion.
Minor points:
-In each figure indicate how many experimental replicates have been made
-lane 179: The reference indicated by the authors describes the HuiPSC MUi009 cell line, not HuiPSC MUi019. Replace with Stem Cell Res. 2017 Apr;20:91-93. doi: 10.1016/j.scr.2017.02.013. Epub 2017 Mar 7. Generation of induced pluripotent stem cells from peripheral blood CD34+ hematopoietic progenitors of a 31year old healthy woman. Tangprasittipap A, Jittorntrum B, Wongkummool W, Kitiyanant N, Tubsuwan A.
-Figure 1: replace G with F in the last panel. Repeat PCR for actin with fewer cycles
-Figure 1G and Figure 3D: please provide the individual immunodecorations (alb., CD81, Dapi).

Reviewer 3 ·

Basic reporting

1. The manuscript is clearly written and is comprehensible. The introduction is sufficient to orient towards the aims and objectives of the work.
2. The raw data supplied is not well-labeled.
3. The resolution of main figures is perhaps low, enlarging the images compromises their quality.

Experimental design

1. The research question is well-stated but given the amount of research going on in this field, more rigorous investigation needs to be performed as elaborated below.
2. Under the methodology section regarding 3D culture of organoids (Line 157-165), the method is not detailed enough. The authors should elaborate how the 3D cultures were maintained for 17 days. Were the organoids passaged? Or was simply the top medium replenished? And how frequently?

Validity of the findings

Major comments:
The study though simple and rapid (as claimed by the authors) suffers from some serious drawbacks, which should be addressed to rightfully claim the overall ingenuity of the work.

1. There is no statistical analysis of the work performed. The authors have not mentioned how many time the organoid culture was repeated which would indicate the reproducibility of the data. There is no mention of the repeats for RT-PCR data analysis or immunofluorescence or glycogen accumulation.
2. One important issue is that the data is only qualitative but not quantitative. This again questions the reproducibility of the data. The authors can perform quantitation of RT-PCR gels and the number of repeats can then be represented graphically or numerically with standard deviation and other statistical information.
3. In the discussion, the authors have claimed the rapidness of their technique by comparing it to that of Mun et al., 2019. It is mentioned that Mun et al achieved spherical structures at day 22 of their culture. I would like to draw attention that the 22 day period in Mun et al paper includes all the stages of differentiation starting from Pluripotent stem cells. I f we take into account the initial stages then for the present paper it will be 10 days for reaching Hepatic endoderm and additional 12 days in 3D culture making it a total of 22 days when they see the spherical structures typical of blastocyst and then require additional 5 days to for maturation. How would you justify the rapidness?
4. Also to justify maturation of organoids the authors rely only on Albumin and Cd81 immunostaining. The characterization of the organoids is not rigorous enough. The authors should do some additional experiments to justify the qualitative efficacy of their protocol as suggested below.
5. Should perform Cyp activity assay or AAT/albumin section in medium to show full functional potential of the hepatocytes.
6. The authors did not show glycogen accumulation in the organoids.
7. There is no comparison of expression data to positive control like human liver tissue or primary hepatocytes to assess/compare the extent of maturation.
8. As can be seen from figure 3D, not all cells in organoid express albumin or CD81, the authors should perform FACS analysis to depict what percentage of organoid cells are mature hepatocytes. This is important if the authors claim to achieve successful generation of liver organoid. It would be great if they are also able to assess the organoids for ductal epithelial cells and endothelial cells using FACs.
9. An important point would be to highlight how the 3D cultured organoids are superior to the 2D differentiated hepatocytes. It can be emphasized by showing a comparison of hepatic and ductal markers in the two along with a positive control such as primary hepatocytes or human liver tissue as well as comparison of functional assays between the two.
10. I know that the supplementary figures are the raw data for the RT-PCR data shown in Figure 1D, I still feel that the supplementary data should be sufficiently labeled, specially the DNA ladder information is missing. The base pair information of individual bands of the ladder must be indicated. Also the expected amplicon size for each primer set should be mentioned either in the primer sequence table or along with raw data.

Additional comments

1. The referencing syntax used is not consistent throughout the text. I think it would be appropriate if all references being mentioned at a particular instance are grouped within single brackets.
2. An important error is the discrepancy in cell-line name. At line 110, the cell line mentioned is MUi009 while in the rest of the text it is referred to as MUi019.
3. The table number is incorrectly indicated in text at line 153 as Table X. I couldn’t find table X in the file.
4. At line 154 in methodology section, it is mentioned that the glass slides were covered with a glass slide. I believe the mounts are normally covered with coverslip, unless it was indeed mounted with another glass slide in this case.

---

## Round 0.2 · accepted · Accept

The revised manuscript is now appropriate for publication in PeerJ. Thank you for your contribution.

Reviewer 2 ·

Basic reporting

no comment

Experimental design

no comment

Validity of the findings

no comment

Additional comments

This version of the manuscript is significantly improved and the authors have addressed all the issues raised in the first revision. I believe that the revised manuscript is appropriate for publication in Peerj.